# Autism Diagnosis Using Iterative Permutation Sampling–Recursive Feature Elimination Algorithm and Deep Learning

Ahmed Shalaby*, Omar Dekhil†, Krishna Kanth Chitta*, Hankyu Lee*, Dwight German‡, Jeon Lee*,

* Lyda Hill Department of Bioinformatics, UT Southwestern Medical Center, Dallas, TX, 75390 USA
† Computer Science and Engineering Department, University of Louisville, KY, 40208, USA
‡Department of Psychiatry, UT Southwestern Medical Center, Dallas, TX, 75390 USA

*Abstract*—**Autism spectrum disorder (ASD) is a neuro-developmental disorder that affects social and communication abilities. There are no confirmed causative factors for the spectrum of symptoms that occur in ASD children. Currently, the gold standard for an ASD diagnosis is based on clinical testing. In particular, brain imaging modalities are believed to hold discriminant information for an ASD diagnosis. Recently, it has been proposed that altered functional connectivity patterns in the resting state functional MRI (RfMRI) coupled with machine learning may hold promise for an ASD diagnosis. However, algorithms that extract these patterns generate a large number of connectivity features, leading to high dimensional data. To address this problem, we propose a novel efficient feature selection algorithm called Iterative Permutation Sampling–Recursive Feature Elimination (IPS–RFE). Only a limited number of informative discriminating features are fed to a deep neural network classifier. We have investigated this approach for classifying ASD in the ABIDE 1 dataset which contains approximately 1000 subjects. The proposed feature selection and classification approach outperforms other state-of-the-art alternatives with an accuracy of 75%, sensitivity of 73.5%, specificity of 76.5% and area under ROC curve of 0.803. A high percentage of the features selected by the IPS-RFE algorithm belong to the default mode, limbic, and visual brain networks, which have been reported to be abnormal among ASD children.**

*Index Terms*—**RfMRI, ASD, IPS-RFE, ABIDE I**

## I. Introduction

Autism spectrum disorder (ASD) is a neuro-developmental disorder associated with impaired social functionality, impaired language abilities, and restricted or repetitive behaviors [1]. Currently, there are no known causes for ASD. ASD is clinically diagnosed using the Autism Diagnosis Observation Schedule (ADOS) [2]. Brain imaging researchers are currently using multiple neuro-imaging modalities to develop more objective methods for ASD diagnosis. The frequently employed modalities include structural MRI (sMRI) for characterizing anatomical and morphological abnormalities [3], diffusion tensor imaging (DTI) for white matter connectivity abnormalities [4], [5], and functional MRI (fMRI) for measuring functional connectivity abnormalities. Two types of fMRI scans have been used for ASD diagnosis; task-based fMRI [6]–[8], and resting state functional MRI (RfMRI) [9]–[11] which we employ in this study.

Many researchers have utilized brain functional connectivities derived from RfMRI to differentiate between ASD and typically developed subjects (TDs) [12]–[16]. In 2014, the Autism Brain Imaging Data Exchange (ABIDE) was formed using data from more than 1000 subjects which was collected from 17 different sites [17]. Demographic details of the ABIDE 1 cohort are provided in the following section (II-A). In [18], the authors utilized features selected from ROI-based RfMRI to identify subjects with ASD. They used a support vector machine (SVM) classifier, achieving a classification accuracy of 67%. Another study [19] combined a denoising autoencoder followed by a multi–layer perceptron neural network for the classification of ASD using functional connectivity features. Although they achieved very high accuracies on each individual site on the ABIDE 1 dataset, a global accuracy they achieved when combining all the sites was 70%. Additionally, the tracking of the original features in the reduced feature space is not an easy task so this also limits the ability to interpret the used features.

A recent study [20] applied volumetric convolutional neural networks to utilize the 3D RfMRI volumes directly. This approach recorded an accuracy of 73% when tested on the ABIDE 1 dataset. Another study [21] utilized features extracted from functional connectivity matrices to identify ASD subjects. They employed various classifiers, with a modified version of a Gaussian radial basis function–SVM achieving the highest classification accuracy of 69.4% for the whole ABIDE 1 dataset. In [22], a subset of the ABIDE 1 dataset was used to extract different graph theoretical features, which were then fed to a Gaussian–SVM classifier. The accuracies reported in this article were divided per age group and varied from 69% to 95%. However, no results were reported for this algorithm when tested on the entire ABIDE 1 cohort. In [23], a subset of the ABIDE 1 dataset only with subject ages less than 20 years old was selected. In this way, the total number of the used subjects in the study was 312 ASDs and 328 TDs. This study applied an empirical threshold to connectivity feature values. These features were then fed to a probabilistic neural network for classification. The accuracy achieved in this study was 90%, but again, there were no reported results for the entire ABIDE 1 dataset. In [24], an accuracy of 91% was reported on a different dataset containing 126 ASDs and 126 TDs, using a random forest (RF) classifier with connectivity features.

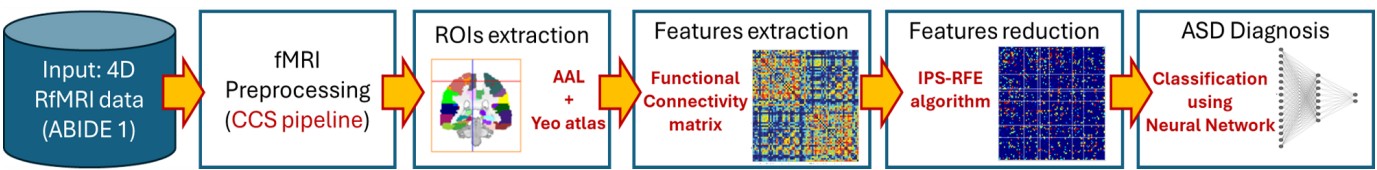

Fig. 1. The framework of the proposed system. The proposed pipeline includes five steps: 1) preprocessing step using the conectome computation system (CCS) pipeline, 2) brain ROIs extraction based on the anatomical atlas labeling (AAL) and Yeo atlases, 3) features extraction for each brain ROI, 4) features reduction using the Iterative Permutation Sampling- Recursive Feature Elimination (IPS-RFE) algorithm, and 5) classification using a deep neural network.

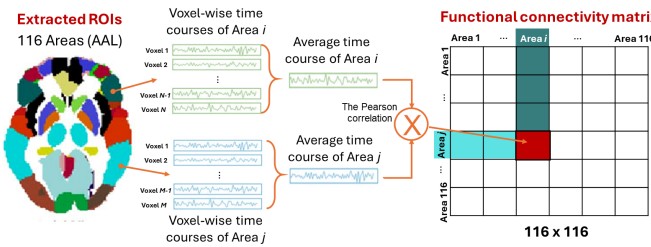

Fig. 2. Illustration of the calculation steps for the functional connectivity matrix using RfMRI and AAL atlas.

One of the major problems facing medical applications of machine learning is the imbalance between the large number of features and the limited number of subjects [25], [26]. This problem leads to overfitting, increased model complexity, and poor generalization of any machine learning model [27], [28]. Due to the difficulty or even impossibility of adding new subjects to the training data, there is usually a need to reduce the number of features instead [29]. In the literature, various techniques have been proposed to reduce the dimensionality of the feature space and they can be grouped into unsupervised or supervised techniques. The review in [30] provides a comprehensive summary of many of these techniques. Authors concluded that for predictive modeling tasks where the goal is to maximize the accuracy of predictions, supervised techniques often provide a significant advantage over unsupervised approaches. To that end, we developed a novel algorithm based on a supervised feature reduction method called recursive feature elimination (RFE). The RFE algorithm [31] was first introduced for selecting genes relevant to cancer classification. The main idea of the RFE is to first fit a model with all the features, rank the feature importance, then start a backward elimination process for the least important features. In the original study, a SVM classifier was used to fit the RFE model. Another variant of RFE was introduced in [32], where a RF classifier was used.

In the current work, we introduce a novel feature selection algorithm for ASD classification that achieves improved accuracy on the ABIDE 1 dataset (Figure 1). Our algorithm is called Iterative Permutation Sampling–Recursive Feature Elimination (IPS–RFE) and followed by a deep neural network for classification of the entire ABIDE 1 dataset. More details about data description and the proposed methods are provided in the following section.

## II. MATERIALS AND METHODS

### A. Data description

The entire cohort of the ABIDE 1 dataset was used in this study*. This dataset was collected from 17 different sites and includes a total of 1,035 subjects with RfMRI scans [33]. The TD group has 95 females and 435 males, while the ASD group has 62 females and 443 males. The proportions of subject genders are not significantly different ($\chi^2 = 6.4, p = 0.11$). Regarding age distribution, the mean age for TDs subjects is $16.8 \pm 7.4$ years. For ASD subjects, the mean age is $17.2 \pm 8.5$ years. The mean ages are also not significantly different ($t = -5.8, p = 0.54$). More information about the scanning parameters for each site are available at the ABIDE dataset website*.

### B. Data preprocessing and ROI extraction

In this study, we used data available from the ABIDE website, which were preprocessed following the connectome computation system (CCS) pipeline [34], consisting of the following steps: (i) removing first four volumes of the input RfMRI scan, (ii) slice–timing correction, (iii) within–subject motion correction, (iv) intensity normalization, and finally, (v) standard space registration. To register the fMRI volumes to the MNI152 standard space, a two stage registration is applied, where the subject is first registered to its high resolution T1-weighted structural scan, then registered to the standard space [34].

The regions of interest (ROIs) in this study are defined according to the anatomical atlas labeling (AAL) [35]. AAL contains 116 anatomical areas. The mean activation time course for each ROI was calculated as the average of the activation time courses of all corresponding voxels in the ROI. For better feature interpretation, a mapping between the AAL atlas and the Yeo functional atlas [36] was performed. The Yeo atlas defines seven human brain functional networks: visual, somatomotor, dorsal attention, ventral attention, limbic, frontoparietal and default mode networks. The mapping was done by assigning a Yeo network label to each AAL area based on the highest number of intersecting voxels between the functional networks and the anatomical areas.

### C. Feature extraction

For each subject, a functional connectivity matrix was calculated. The functional connectivity between a pair of AAL ROIs

*http://fcon_1000.projects.nitrc.org/indi/abide/abide_I.html

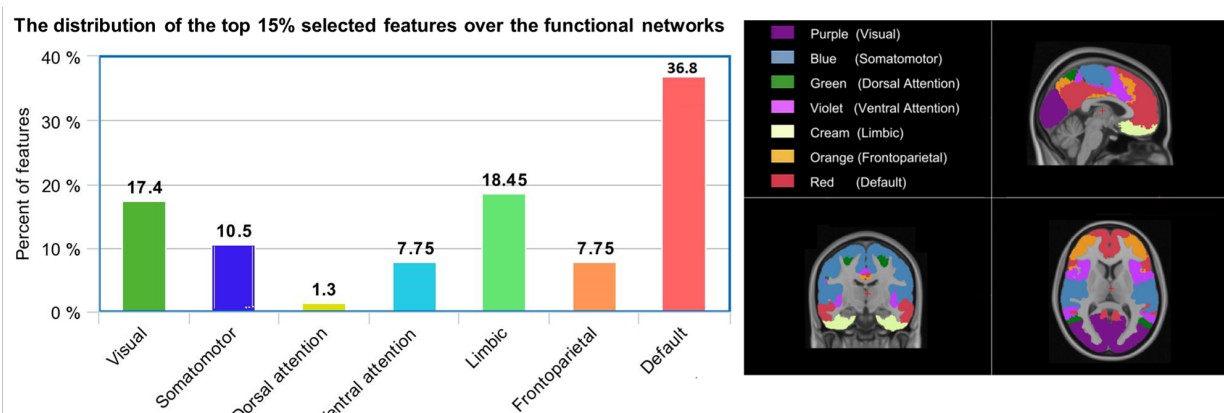

Fig. 3. The distribution of the selected features over the seven functional networks (left panel), and the cross sectional view of the seven functional networks (right panel) based on Yeo functional atlas.

in the brain is defined as the Pearson correlation coefficient between the average time courses of these two ROIs (Figure 2). The connectivity matrix is of size 116x116. The dimensionality of the features after removing symmetric redundancies (i.e. after discarding the upper triangle and the diagonal elements of the output matrix) is given by $n = \frac{N*(N-1)}{2}$ , where $N = 116$, which gives $n = 6670$. The number of features is much higher than the number of subjects, which introduces complications in the learning task. To solve this problem, we propose the IPS-RFE algorithm.

In this study, an RF classifier was used to first fit the RFE model. RF is an ensemble machine learning algorithm combining multiple decision trees using bootstrap aggregation [37]. Each decision tree is fed with a bootstrap of the data. In order to perform feature selection with an RF model, GINI impurity was adopted. The GINI impurity is used to define the probability of incorrect classification of a new instance of the data if this instance is classified according to the distribution of the labels in the used subset in the training phase [38]. It is given by:

$$I_G = 1 - \sum_{j=1}^{c} p_j^2 \qquad (1)$$

Where $p_j$ is the probability of each of the $c$ classes. The RFE output is a binary vector of the same length as the feature vector to indicate whether the feature should be included or not. However, applying RFE to the whole data leads to two main problems: (i) overfitting the features on the used data which reduces the generalization ability, and (ii) extracting false–positive, noisy features that should not be included in the selection. To overcome such problems, we combine RFE with an iterative permutation sampling (IPS) algorithm.

As shown in Algorithm 1, each iteration in the IPS–RFE algorithm is summarized into the following steps: (i) creating a vector of zeros of the same length as the features vector, called the accumulator which will hold the number of times a feature is selected over the iterations, (ii) shuffling the data, (iii) randomly sampling a fraction $f$ of the data, where $f$ is set to 0.8 in our case, (iv) running the RFE using a $K$-fold cross validation on the selected partition, (v) for each selected

feature, incrementing its count in the accumulator by one, and (vi) thresholding over the accumulator value to select the top $p$ percentile of the features, where $p$ is empirically selected to be 15% in this study.

---

**Algorithm 1** IPS-RFE feature selection algorithm

**Result:** The accumlator array that show how many times a feature was selected

//initialization
$n\_features = 6670$
$accumaltor = zeros(n\_features, 1)$
$counter = 0$
**while** *counter <1000* **do**
   1 - Randomly shuffle the data
   2- Select 0.8 of the data
   3- Run the RFE using random forest on the selected data
   4- $RFE\_output$ = Extract the prominent features of the selected data
   5- $i = 0$
   6- **while** $i <n\_features$ **do**
      **if** *RFE_output[i] == True* **then**
         $accumaltor[i] = accumaltor[i] + 1$
      **end**
      $i ++$
   **end**
   $counter ++$
**end**
$return\ accumaltor$

---

### D. Classification of the selected features

Using the top selected features, a deep neural network was then used for the classification task. To find the optimal set of hyperparameters for the neural network, a heuristic grid search was applied. The hyperparameters are: (i) the number of layers in the network, (ii) the number of nodes in each layer, (iii) the $L2$ regularization term, and (iv) the learning rate. To verify statistical significance of the proposed classifier, a permutation test was conducted. More details about the structure of the adopted network and its hyperparamters are shown in section III. Here, the classification labels were randomly shuffled to simulate a random uninformative dataset. This shuffling was repeated 1000 times, and each time it was fed to the classifier to evaluate the classification accuracy.

## III. EXPERIMENTAL RESULTS

### A. IPS-RFE results

In this experiment, the top 15% of the features in the accumulator vector were selected. The number of selected features was 1002. Figure 3 shows the distribution of the selected features over the seven functional networks. In addition, it shows the coronal, axial, and sagittal cross section views of the functional networks. From this figure, it is apparent that the functional networks with the most number of selected features are the default mode network, the limbic network, and the visual network. Figure 4 shows a heat map of the correlation matrix of the AAL atlas ROIs with corresponding labels to the seven Yeo functional networks. This color coded map shows the number of times a feature was selected over the 1000 iterations. For better visualization, only the 1002 selected features are displayed and the rest of features are mapped to dark blue.

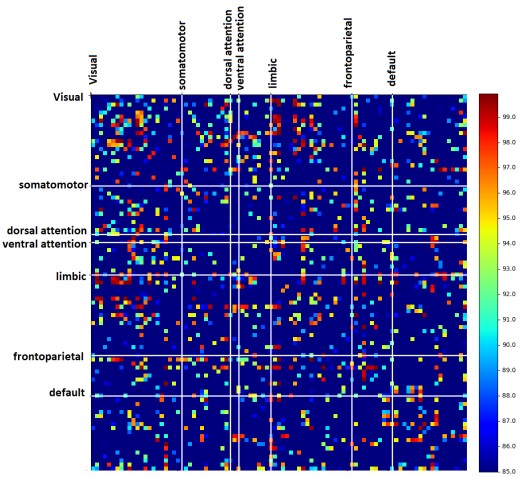

Fig. 4. A color coded representation of the number of times a feature was selected over the 1000 iterations. Only the top $15^{th}$ percentiles features are displayed and the rest of features are mapped to dark blue.

Previously reported findings in the literatures about the altered connectivity patterns of autistic subjects relate to both the visual and default networks. In [24], which analyzed 302 subjects from the ABIDE 1 dataset, the three main networks containing discriminant features were the default mode, visual, and somatosensory networks, which overlap with our findings. Impaired connectivities of the default mode network were also reported in [39], where a statistically significant finding was reported on a data set of 57 ASD and 57 TD subjects. In [40], the visual network showed significance in both statistical testing and classification ability when using a dataset of 817 subjects in the ABIDE I dataset.

In order to study the separability of the data based on our proposed feature selection algorithm, the t-SNE dimensionality reduction technique was utilized [41]. T-SNE allows for visualization of high dimensional data through projection on a 2D plane. Moreover, to quantitatively evaluate the class separation, we calculated the Silhouette Score (SC) [42] between ASDs and TDs classes in the t-SNE space. This metric measures

TABLE I
THE ACHIEVED ACCURACY, SENSITIVITY, SPECIFICITY AND AREA UNDER ROC. ON THE RIGHT COLUMN THE CONFUSION MATRIX IS DISPLAYED.

| Accuracy | 74.9% |
|---|---|
| Sensitivity | 73.5% |
| Specificity | 76.4% |
| AUC | 0.803 |

Confusion matrix

|  | ASD | TD |
|---|---|---|
| ASD | 0.735 | 0.265 |
| TD | 0.236 | 0.764 |

TABLE II
THE SIX MODEL HYPERPARAMETERS USED IN THE GRID SEARCH FINE TUNING, THEIR SEARCH RANGES AND THE SELECTED VALUES.

|  | Hyperparameter | Search range | Selected value |
|---|---|---|---|
| RFE | RFE number of trees | 100-1000 | 500 |
|  | Maximum depth per tree | 2-100 | 20 |
| DL | Neural network number of hidden layers | 1-5 | 2 |
|  | Number of nodes per layers | 5-1000 | 499 for layer 1 150 for layer 2 |
|  | L2 regularization parameter | 0.000001-0.001 | 0.000489 |
|  | Learning rate | 0.00001-0.01 | 0.001 |

how similar an object is to its own class compared to other classes. The SC ranges from -1 to 1, where higher values indicate better-defined clusters. Figure 5 shows the two class data points when using all features (mean SC = -0.0132), the top 50% of the features (mean SC = -0.0038), the top 25% of the features (mean SC = 0.0241), the top 15% of the features (mean SC = 0.0820), the top 10% of the features (mean SC = 0.0192), and the top 5% of the features (mean SC = -0.0557). It is visually and quantitatively apparent that using the top 15% of the input features increases the class separability of the data points.

### B. Classification results

In this study, we used a 4-fold cross validation for classification evaluation. The calculated metrics were accuracy, sensitivity, and specificity. To evaluate the classifier robustness, we also calculated the area under the receiver operating characteristic (ROC) curve and generated a confusion matrix. Table I lists these metrics. To find the optimal set of hyperparameters, a heuristic grid search was performed. For fitting the RFE model with the RF classifier, the two hyperparameters to be tuned were the number of trees and the maximum depth per tree. The ADAM optimizer [43] was used for the neural network. Table II presents the search ranges of each hyperparameter and the selected value.

To show that our system outperforms other state-of-the-art techniques reported using the same dataset with the same experimental design and validation techniques, Table III presents the accuracy of the proposed system and the accuracies reported in the literature [19]–[21], [44]. Our proposed system outperforms other alternatives with an accuracy of 75%, sensitivity of 73.5%, specificity of 76.5% and area under ROC curve of 0.803.

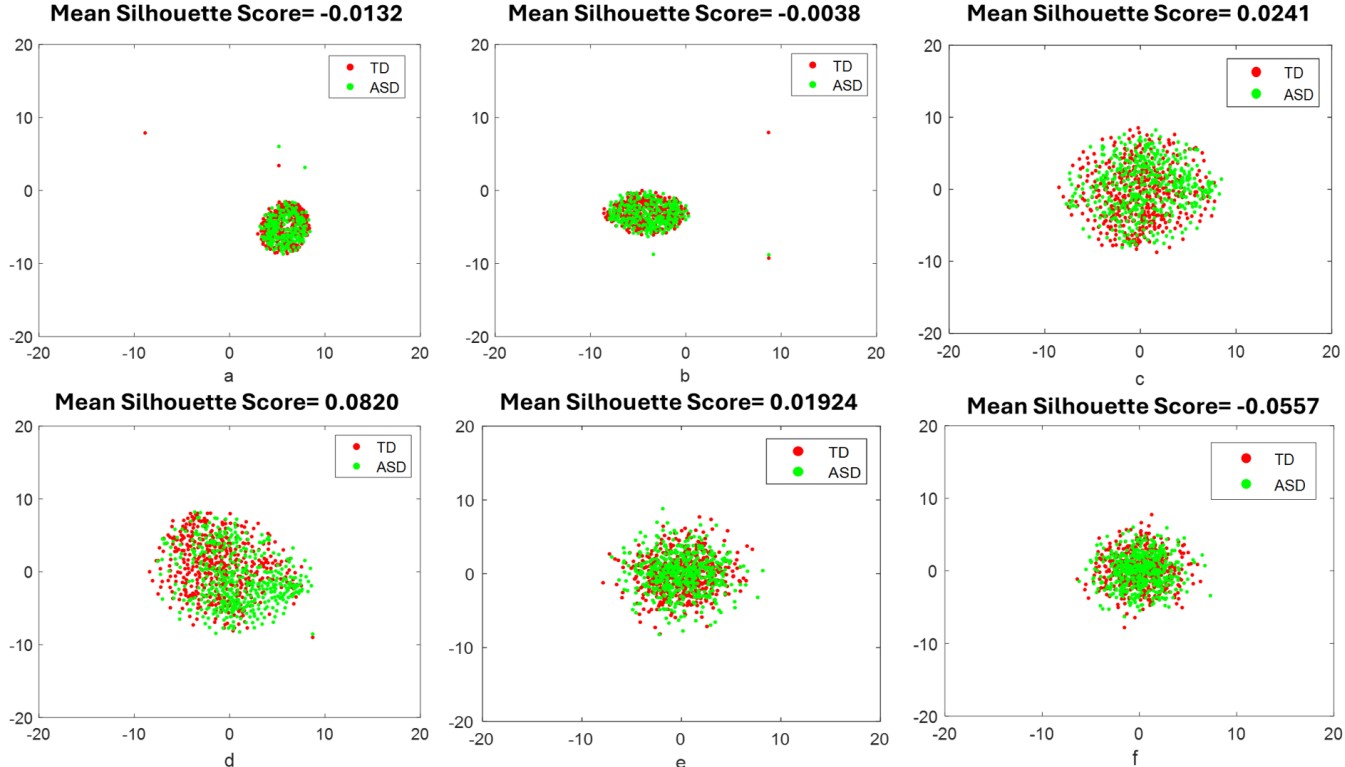

Fig. 5. The t-SNE visualization and the mean silhouette score of the data when: (a) using all the features, (b) using the top 50% of the features, (c) using the top 25% of the features, (d) using the top 15% of the features, (e) using the top 10% of the features, and (f) using the top 5% of the features. It is obvious that by selecting the top 15% of the features, the data becomes more separable (highest mean SC).

TABLE III
A COMPARISON BETWEEN OUR PROPOSED APPROACH AND THE TOP 4
STATE-OF-THE-ART APPROACHES REPORTED IN THE LITERATURE.
ACCUARCY , SENSITIVITY AND SPECIFICITY ARE IN %

| Research | Algorithm | Acc. | Sens. | Spec. | AUC |
|---|---|---|---|---|---|
| Heinsfeld et al. [19] | Denoising autoencoder + NN | 70.0 | **74** | 63 | 0.73 |
| Chaitra et al. [44] | Recursive-clustering SVM | 70.1 | – | – | – |
| Khosla et al. [20] | Convolutional neural network | 73.3 | 72.5 | 74.2 | 0.75 |
| Yang et al. [21] | Kernel-based SVM | 69.4 | 64.6 | 73.6 | 0.75 |
| **Proposed approach** | **IPS-RFE + Deep NN** | **74.9** | 73.5 | **76.5** | **0.80** |

On the other hand, the mean accuracy of a permutation test by creating 1000 iterations of random label shuffling and feeding them to the classification model is 52%. It can be inferred that the proposed system significantly performs better than random guessing. This also shows evidence that there are no sources of data leakage between training and testing data. Such leakage will lead to higher accuracies with random label shuffling, which is not the case in this study.

To study sex-based differences in the prevalence of ASD, we evaluated the performance of the proposed system using only the males scans from ABIDE 1 dataset (in total 878 subjects) and compared it with the performance of our system when the whole dataset is used, as shown in Figure 6. The accuracy for males only was 76%, sensitivity = 74.1%, specificity = 77.7%, and area under ROC curve = 0.819. These results indicated

that the performance of the proposed system is nearly identical whether using only male scans or both male and female scans, likely due to the significantly lower number of female subjects in this dataset.

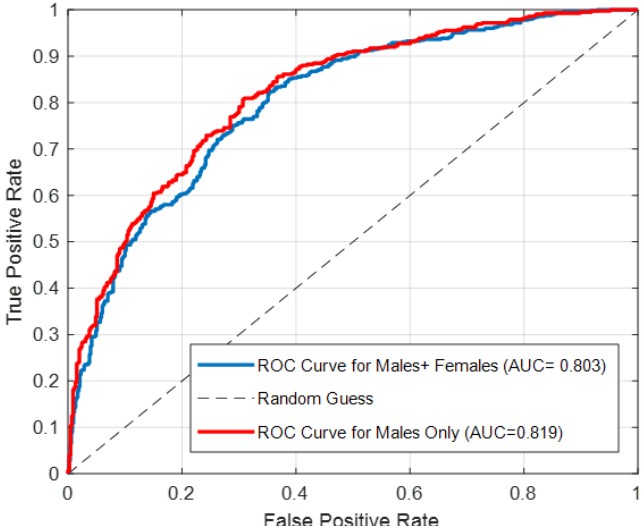

Fig. 6. The ROC curve (with the AUC values) comparison illustrating the classification performance difference between using data from males only versus combining data from males and females.

## IV. Conclusion and Future work

In this work, we propose a feature selection and classification system for autism diagnosis using functional connectivity features extracted from RfMRI. The dataset used in this study is the ABIDE 1 dataset with 1035 subjects. The main idea of the proposed system is to select the most significant features in identifying autistic subjects from typically developed peers. Due to the high number of features, it is crucial to find a way for denoising the features by keeping discriminant ones. This is done by combining an iterative permutation sampling algorithm with a recursive feature elimination algorithm. Subsequently, only 1002 significant features are fed into a deep neural network for ASD diagnosis. We evaluated the performance of the proposed system by assessing the classification accuracy. The proposed system outperforms other alternatives, achieving a classification accuracy of 75%. In addition, we also provide a map with the most prominent features. This map is defined on the AAL atlas and also mapped to the Yeo atlas which defines seven functional networks. We show that the default mode network, the limbic network, and the visual network contain the most number of discriminating features, which agrees well with previous literatures [24], [40].

Although this model outperforms other state-of-the-art models in overall accuracy for the whole ABIDE 1 RfMRI dataset, we believe that there is room for more improvements. One possible future direction is to adopt a multi-modal approach to further improve classification accuracy [38]. This approach might include modalities such as the transcriptome, DTI, and structural MRI. Combining evidence from these MRI modalities with genomic data and early clinical behavioral assessments in larger sample sizes would allow for more precise quantifications of ASD subgroups.

Despite the widely recognized sex-based differences in ASD [45], we could not find any performance differences between the males and females in our experiments likely due to the significantly lower number of female subjects in the ABIDE 1 dataset. To deeply study the sex-based differences in ASD using RfMRI, a more balanced dataset with a higher number of female subjects is needed. Another approach is to consider how these systems could be integrated with personalized medicine. This means that the output of the system will not only be a binary prediction on whether the subject is autistic or not, but will also include a detailed report about the affected areas of the brain and to what extent they are affected. This would allow physicians to design a personalized treatment plan depending on each individual patient's needs.

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
