# OpenReview forum: "Autism Diagnosis using Iterative Permutation Sampling-Recursive Feature Elimination Algorithm and Deep Learning"
_IEEE.org/EMBS/BHI/2024/Conference — IEEE BHI'24_

### Official Review · Reviewer_5wxL · 2024-08-06
**Reviewer comments  for Paper ID 340**

**Overall Rating:** 8
**Confidence:** 5

**Other Quality Metrics:**

(a) Clarity of Writing - Very well done.
(b) Clinical Significance - Not applicable.
(c) Methodological Novelty - Significant.
(d) Experiments and Results - Very good.

**Questions For The Authors:**

The paper is well-written, with a clearly formulated research question. The claims made are well-justified by the results provided.

**Strengths:**

1. The paper is well-written and well-formulated.
2. The figures are of high quality.
3. The results and discussion are well-presented.
4. The abstract is well-constructed.

**Summary Of The Paper:**

The authors have proposed a novel and efficient feature selection method called Iterative Permutation Sampling-Recursive Feature Elimination for deep learning in autism diagnosis. They claim that this new method outperforms the state-of-the-art approaches by achieving a 75% improvement in accuracy.

**Weaknesses:**

Nothing to mention

---

### Official Review · Reviewer_QeHR · 2024-08-10
**Autism Diagnosis using Iterative Permutation Sampling-Recursive Feature Elimination Algorithm and Deep Learning: A Review**

**Overall Rating:** 6
**Confidence:** 3

**Other Quality Metrics:**

1. Clarity of Writing: Good
2. Clinical Significance: Good
3. Methodological Novelty: Great
4. Experiments and Results: Good

**Questions For The Authors:**

1. Feature Selection Clarification: Could you elaborate on how the threshold for selecting the top 15% of features was empirically determined? Would different thresholds significantly affect the model's performance?
2. Gender Differences: Given the acknowledged gender imbalance in the dataset, how do you anticipate the model would perform with a more balanced dataset?
3. Clinical Applicability: How do you envision the IPS-RFE algorithm being used in a clinical setting? What additional validation steps would be necessary before this approach could be applied to clinical diagnostics?

**Strengths:**

1. Innovative Algorithm: The development and implementation of the IPS-RFE algorithm represent a significant contribution to the field, addressing the challenge of high-dimensional data in brain imaging studies.
2. Application of Deep Learning: The use of a deep neural network for classification adds robustness to the model and helps achieve relatively high accuracy compared to other methods.
3. Comprehensive Evaluation: The study rigorously evaluates the model's performance, including accuracy, sensitivity, specificity, and area under the ROC curve, providing a clear understanding of its effectiveness.
4. Contextualization of Results: The paper successfully places its findings in the context of existing literature, identifying overlaps with known abnormalities in ASD-related brain networks.

**Summary Of The Paper:**

This paper presents a novel approach to diagnosing Autism Spectrum Disorder (ASD) using resting-state functional MRI (RfMRI) data. The authors propose a new feature selection algorithm, Iterative Permutation Sampling-Recursive Feature Elimination (IPS-RFE), to handle the high-dimensional nature of the data, followed by a deep neural network for classification. This approach is applied to the ABIDE 1 dataset, which consists of approximately 1,000 subjects. The proposed method achieves a classification accuracy of 75%, with a sensitivity of 73.5% and specificity of 76.5%. The study identifies significant features related to the default mode, limbic, and visual networks, which are known to be associated with ASD.

**Weaknesses:**

1. Limited Dataset Diversity: The ABIDE 1 dataset, while extensive, is limited by its demographic composition, particularly the imbalance between male and female subjects. This limitation might affect the generalizability of the model.
2. Interpretability of the Model: While the IPS-RFE algorithm effectively reduces the feature space, the interpretability of the selected features in terms of their biological relevance could be further explored.

---

### Official Review · Reviewer_6Py7 · 2024-08-26
**Review of Submission 340**

**Overall Rating:** 6
**Confidence:** 4

**Other Quality Metrics:**

(a) Clarity of writing: Good;
(b) Clinical Significance: Good;
(c) Methodological Novelty: Fair;
(d) Experiments and Results: Fair

**Questions For The Authors:**

Please check the Weaknesses.

**Strengths:**

1. The scope of this paper is an important topic. Automated detection of ASD is gaining much attention in the last few years. The authors provided a good overview of prior work in their literature review.

2. The methodology section is very well-explained.

**Summary Of The Paper:**

In this paper, the authors proposed a novel feature selection method (Iterative Permutation Sampling–Recursive Feature Elimination (IPS–RFE)) for deep learning based Autism spectrum disorder (ASD) detection using resting state fMRI (RfMRI) data. The proposed method reduces the large feature space generated by functional connectivity matrix using Random Forest based recursive feature elimination method that minimizes the gini index to reduce the misclassification. Their feature selection results in a significantly lower feature space (15%) for the following deep learning models that resulted in ~75% accuracy.

**Weaknesses:**

1. As the dataset is very imbalanced, comparing only the accuracy with prior work might not be a good choice.  Comparison also should have been done using other metrics (AUC, sensitivity, specificity) which they already measured for the proposed method. How are these metrics different in prior work from the proposed method?

2. The choice of using 15% features seemed empirical. Was there any reasoning behind this number? The t-SNE plots in Fig.5(c) does not show good separation of classes for the dataset using the 15% selected features. What happens if this number decreases? Does that increase separation or improve performance?

---

### Decision · Program_Chairs · 2024-09-23

Accept